# Lymphangiogenesis in the Deepest Invasive Areas of Human Early-Stage Colorectal Cancer

**DOI:** 10.3390/ijms26072919

**Published:** 2025-03-24

**Authors:** Masaharu Tanaka, Qian Zhou, Minako Ohnishi, Miho Kandori, Ami Itou, Yuki Kitadai, Hidehiko Takigawa, Shiro Oka, Akiko Kimoto, Fumio Shimamoto, Yasuhiko Kitadai

**Affiliations:** 1Department of Health Sciences, Faculty of Human Culture and Science, Prefectural University of Hiroshima, Hiroshima 734-8558, Japan; miyabi.hinata1116@gmail.com (M.T.);; 2Graduate School Biomedical and Health Science, Hiroshima University, Hiroshima 734-8551, Japan; 3Department of Gastroenterology, Hiroshima University Hospital, Hiroshima 734-8551, Japan; 4Faculty of Health Sciences, Hiroshima Cosmopolitan University, Hiroshima 734-0014, Japan

**Keywords:** tumor, TAMs, colorectal cancer, angiogenesis, lymphatic vessels

## Abstract

Tumor-associated macrophages (TAMs) are known to induce epithelial–mesenchymal transition (EMT) and angiogenesis in areas with a high density of accumulation in the submucosal (SM) layer. However, lymphatic vessels, which are important routes for lymph node metastasis, have rarely been analyzed, and their relationship to TAM accumulation is unknown. In this study, paraffin-embedded sections from 11 cases of human early-stage colorectal cancer (SM invasive carcinoma) were stained with CD34 antibody for vascular endothelium and podoplanin antibody for lymphatic endothelium at the deepest, central, and marginal sites of tumor invasion. Tumor blood vessels increased in the deepest invasive areas, and a positive correlation was observed between the number of TAMs and tumor blood vessels. Interestingly, lymphatic vessels with CD34-positive endothelial cells (CD34-positive lymphatic vessels) were observed within the tumor. The number of CD34-positive lymphatic vessels was significantly higher in the metastasis-positive group. These results suggest that abnormalities in the vascular and lymphatic systems are observed from the early stage of colorectal cancer development and that VEGF-A derived from TAMs is important for tumor angiogenesis. In addition, CD34-positive lymphatic vessels observed in the deepest areas of tumor invasion have not been reported in Japan, with initial reports indicating that they are neoplastic lymphatic vessels.

## 1. Introduction

According to global statistics collected in 2020 on cancer by organ, colorectal cancer ranked third and second in age-adjusted morbidity and mortality for men and women, respectively, and third and fourth in age-adjusted mortality for men and women. These data highlight that colorectal cancer is a highly malignant cancer, ranking highly in both sexes. At present, our laboratory is conducting research with the aim of reducing the morbidity and mortality associated with colorectal cancer [1].

Colorectal cancer begins in colorectal mucosa and grows locally, spreading throughout the body via invasion and metastasis. The tumor microenvironment plays a major role in cancer metastasis. Within the tumor microenvironment, stromal, endothelial, and immune cells are localized [2]. Macrophages in the tumor microenvironment, also known as tumor-associated macrophages (TAMs), are abundant and play a role in cancer metastasis, which is a multi-step process. Cancer involves suppressing tumor immunity, leading to the proliferation, invasion, and migration of cancer cells, which in turn promotes angiogenesis. Metastasis is established when cancer cells invade the neovascular vessels generated during this process and migrate and proliferate in other organs. All of these steps must be overcome for cancer metastasis to occur. The liver is the main organ to which colorectal cancer metastasizes. This is based on Paget’s seed and soil theory, which states that the metastatic site is determined by the organ’s microenvironment, which is similar to that of the primary tumor [3]. There are three major metastatic routes: hematogenous metastasis occurs when cancer cells invade blood vessels, lymphatic metastasis occurs when cancer cells invade lymphatic vessels, and peritoneal dissemination [4].

Hematogenous metastasis involves metastasis to organs such as the liver and lungs via the bloodstream, while lymphatic metastasis involves metastasis to lymph nodes via neoplastic lymph vessels. (1) The angiogenic factor VEGF-A is an important angiogenic factor in human colorectal cancer [5]; (2) VEGF-A is produced not only by cancer cells but also by macrophages; and (3) VEGF-C/D is a lymphangiogenic factor in gastric and colorectal cancer [6,7,8].

In our previous studies, we found that various factors in the tumor microenvironment are involved in metastasis. Elucidating their mechanisms in cancer and the microenvironment provides an opportunity to gain insights useful for the treatment of colorectal cancer. Therefore, we focused our research on the angiogenesis and lymphangiogenesis phases. Angiogenesis is initiated by tumor growth, the mobilization of immune cells, and the expression of factors that promote neoplasia. Here, we focused on VEGF, an endothelial growth factor produced by cancer cells and TAMs, which activates receptors expressed on endothelial cells. VEGF-A activates its receptor on endothelial cells, which in turn activates the vascular endothelial cells to proliferate, migrate, and invade through a variety of signaling pathways. Recently, we found that high-density accumulation areas of TAM in the deepest colorectal cancer invasion areas are associated with the first steps in metastasis [9], such as the induction of epithelial–mesenchymal transition (EMT) and the promotion of angiogenesis. However, the study of angiogenesis and TAM remains an open question owing to the small sample size and the lack of discussion on lymphangiogenesis in the deepest tumor zone. The present study was conducted to elucidate the expression of angiogenesis in areas of high-density accumulation of TAM and the mechanism of lymphangiogenesis.

## 2. Results

### 2.1. High-Density Accumulation Area of TAM in the Deepest Tumor Zone

The distribution of TAMs in the deepest tumor infiltration zone was examined using fluorescent double staining (CD68: macrophages, CD34: vascular endothelial cells). As a result, areas of TAM were found to align with the cancer epithelium at the invasive tip and area A, as well as areas of TAM clustering (area B) at a depth of 200–500 µm from the deepest invasive site (Figure 1).

### 2.2. Characteristics of TAMs and Tumor Blood Vessels

To examine the heterogeneity of macrophage distribution within the tumor, TAMs and vascular endothelial cells were fluorescently immunostained. More TAMs and vascular endothelial cells were found in the deepest infiltration zone than in the tumor center (Figure 2).

### 2.3. Correlation Between TAMs and Tumor Vessel Count

The number of CD68- and CD34-positive cells per 1.38 × 10^5^ μm^2^ in the deepest infiltration zone and tumor center, respectively, was measured at multiple locations in each case, and the mean values were determined. Analysis of the correlation between the two showed a significant positive correlation between them. As shown in the graph, the number of TAMs and vascular endothelial cells was found to be higher in the deepest invasive area than in the tumor center (Figure 3).

For each case, the number of CD68- and CD34-positive cells per 1.38 × 10^5^ μm^2^ were measured at multiple locations in the deepest infiltration zone and tumor center, respectively, and the mean values were calculated. Analysis of the correlation between the two showed a significant positive correlation. In other words, a significant positive correlation was found between the number of TAMs and the number of tumor micro-vessels. Compared to the tumor center, the TAM accumulation sites in the deepest infiltration zone were found to have abundant blood vessels.

### 2.4. TAM, VEGF-A, and Tumor Vascular Characteristics

Fluorescent immunostaining of TAMs and VEGF-A showed a high number of cancer cells and TAMs (Figure 4). However, as previously mentioned, the results of fluorescent immunostaining of TAMs and vascular endothelial cells showed that vascular density increased only around TAMs (Figure 4).

### 2.5. Characteristics of Tumor Lymphatic Vessels

The lymphatic vessels in the deepest tumor invasion area were characterized by dilation compared to those in the central and superficial areas of the tumor (Figure 5).

### 2.6. Number of Tumor Lymphatic Vessels

The number of lymphatic vessels per 1.38 × 10^5^ μm^2^ was measured at three locations in each of the tumors (deepest, central, and surface areas). The results revealed that the average number of lymphatic vessels in the deepest tumor invasion area was significantly higher than in the central and surface areas (Figure 6, *p* < 0.01).

The results revealed that the average number of lymphatic vessels in the deepest infiltration area was significantly higher than in the central and surface areas.

### 2.7. Fluorescent Immunostaining for CD34 and Podoplanin

CD34- and podoplanin-positive vessels (hereafter referred to as CD34-positive lymphatic vessels) were frequently found in the deepest infiltration zone. CD34-positive lymphatic vessels were found not only in the deepest infiltrated zone but also in the tumor center. Conversely, few CD34-positive lymphatic vessels were found in normal areas (Figure 7).

### 2.8. Association with Lymph Node Metastasis

No association was observed between the number of lymphatic vessels in the deepest tumor zone and lymph node metastasis (Figure 8). By contrast, an association was found between the number of CD34-positive lymphatic vessels and lymph node metastasis in the deepest tumor infiltration zone (Figure 9, *p* < 0.05). These results indicate that tumors with lymph node metastasis have a significantly higher number of CD34-positive lymphatic vessels than tumors without lymph node metastasis.

The mean number of chimeric and lymphatic vessels in the deepest invasive area was analyzed to determine if there is an association with lymph node metastasis. The results showed no association between the number of lymphatic vessels and lymph node metastasis in the deepest invasive zone. However, an association was observed between the number of CD34-positive lymphatic vessels and lymph node metastasis. These findings indicate that tumors with lymph node metastases had a significantly higher number of CD34-positive lymphatic vessels than tumors without lymph node metastases.

## 3. Discussion

In recent years, the percentage of early-stage colorectal cancer cases has been increasing owing to the development of endoscopic examination equipment and treatment techniques. While metastasis is extremely rare in cancers that remain within the mucosa, when cancer cells invade the submucosa, interaction with stromal cells results in changes to the microenvironment around the tumor, and distant metastasis such as lymph node metastasis may occur. If the cancer is confined to the mucosa, no metastasis is observed. However, if the cancer invades the submucosa, the risk of metastasis increases. In addition, while colorectal cancer remains within the mucosa, angiogenesis is minimal, but as it invades the submucosa, microvascular density increases predominantly. Therefore, in the deepest invasive zone of the colorectal submucosal (SM) layer, the histology of the zone and microvessel density are important factors for lymph node metastasis. Although vascular density becomes heterogeneous as the tumor progresses, it is generally higher in the margins of the tumor than in the superficial and central areas of the tumor; TAM and tumor vessels increase in the deepest areas compared to those in the tumor center, and there is a significant positive correlation between the number of TAM and tumor vessels. In addition, VEGF expression was observed in both cancer cells and TAM, while vascular density increased only in the vicinity of TAM. This suggests that TAM-derived VEGFA promotes angiogenesis and increases vascular density in the deepest tumor zone.

Lymphangiogenesis was also observed more frequently in the deepest areas of tumor invasion, suggesting that lymphangiogenesis occurs in these areas. CD34-positive lymphatic vessels were not identified in normal tissues but were found in cancer tissues, suggesting that CD34-positive lymphatic vessels are neoplastic lymph vessels. However, although no significant difference was observed between the presence of lymph node metastasis and the number of lymph vessels, a significant difference was observed between the number of CD34-positive lymph vessels and lymph nodes. These findings suggest that neoplastic lymphatic vessels, rather than existing lymphatic vessels, may induce lymph node metastasis.

## 4. Materials and Methods

In this study, paraffin-embedded sections of tissues from 11 cases of human early-stage colorectal cancer (SM invasive carcinoma) were stained with CD34 antibody for vascular endothelium and podoplanin antibody for lymphatic endothelium at the deepest, central, and marginal sites of tumor invasion. The staining results were quantified using image analysis software.

### 4.1. Study Population

Eleven randomly selected patients with colorectal neoplasia treated at Hiroshima University Hospital (Hiroshima, Japan) were enrolled in this study. The use of clinical specimens and patient data was approved by the Institutional Review Board of Hiroshima University. This study was performed in accordance with the ethical standards of the 1964 Declaration of Helsinki and its subsequent amendments.

### 4.2. Histological Evaluation

All pathological characteristics of the specimens were evaluated by pathologists (Y.K. A.K. F.S.) according to the latest World Health Organization classification. A gastrointestinal pathologist (F.S.) performed a detailed re-evaluation and diagnosis of the specimens of SM invasive cancer, specifically the pathological findings at the deepest invasive portion, according to our previous report.

### 4.3. Assessment of the Distribution, Phenotype, and Number of TAMs

TAM polarization and distribution were assessed using double immunofluorescence staining with CD68 (a marker of pan-macrophages) and CD163 (a marker of M2 macrophages). M1 macrophages were defined as CD68^+^ and CD163^−^, and M2 macrophages were defined as CD68^+^ and CD163^+^. The distribution of TAM was evaluated by dividing the tumor tissues into four parts (center, lateral periphery, deepest invasive portion, and non-neoplastic mucosa adjacent to the tumor), in which the number and phenotype of the macrophages were evaluated in each section (Figure 1). For evaluation, all slides were reviewed under a fluorescence microscope (BZ-X710; Keyence, Osaka, Japan). For each sample, five high-power fields (HPFs; 1 HPF = 0.0988 mm^2^) showing substantial macrophage infiltration were selected and photographed. The number of M1 and M2 macrophages in each captured image was counted using an image analyzer (WinRoof software 2023 standard; Olympus, Tokyo, Japan) (Table 1).

### 4.4. Immunofluorescence Staining

Double immunofluorescence staining was performed using the Opal 4-colour manual IHC kit (NEL810001KT; PerkinElmer, Waltham, MA, USA). All specimens were fixed in 10% formaldehyde and embedded in paraffin according to routine procedures at the Department of Clinical Pathology, Hiroshima Universal Hospital. One 4 µm section from each specimen was cut and deparaffinized, followed by heat-induced antigen retrieval for 15 min using a microwave after pre-treatment of tissues with 0.3% H_2_O_2_. The slides were then incubated with an anti-CD34 antibody (413111; Nichirei Bio Science, Danvers, MA, USA) for 1 h at 37 °C. Next, the slides were washed and incubated with the secondary antibodies included in the kit for 15 min at 37 °C, followed by washing and incubation at room temperature for 10 min with the Opal570 reagent (Osaka, Japan) included in the kit. After another round of heat-induced antigen retrieval, the slides were incubated with an anti-human Podoplanin antibody (1:400; Angio Bio Co., Ltd.; Leica Biosystems, Newcastle, UK) overnight at 4 °C. The slides were then washed and incubated with the secondary antibodies included in the kit for 15 min at 37 °C, followed by washing and incubation at room temperature for 10 min with the Opal520 reagent included in the kit. After washing, the slides were counterstained with 4′,6-diamidino-2-phenylindole (DAPI) (1:500) for 5 min and mounted.

### 4.5. IHC Staining

IHC staining was performed using an EnVision^TM^ kit (K5007; Dako, Glostrup, Denmark). After the deparaffinization of the tissues, heat-induced antigen retrieval was performed for 15 min, followed by incubation with the primary antibodies anti-E-cadherin (1:100; sc7870; Santa Cruz Biotechnology, Dallas, TX, USA) for 1 h at 37 °C after pre-treatment of tissues with 0.3% H_2_O_2_. The slides were then washed and incubated with the secondary antibodies included in the kit for 30 min at 37 °C, after which the slides were washed and incubated for 7 min at room temperature with the 3,3′-diaminobenzidine included in the kit. The slides were counterstained with hematoxylin; the slides were mounted after dehydration and attainment of transparency.

### 4.6. Statistical Analysis

Quantitative data are presented as the mean ± standard deviation or as percentages. The Shapiro–Wilk test was used to determine whether datasets were normally distributed. Between-group differences were evaluated using Student’s *t*-test or the Mann–Whitney *U* test for quantitative data and a chi-squared test or Fisher’s exact test for categorical data. For multiple comparisons, a one-way analysis of variance, a Kruskal–Wallis test, or the chi-squared test with a Bonferroni post-hoc test was used, as appropriate. All tests were two-sided, and a *p* < 0.05 was considered statistically significant, whereas a *p* < 0.0167 or a *p* < 0.0083 was considered statistically significant for multiple comparisons. Receiver operating characteristic curve analysis with maximal Youden index values was used to identify optimal cut-off values and evaluate lymph node metastasis prediction. We used JMP Pro statistical software (v.14.0.0; SAS Institute, Cary, NC, USA) for all statistical analyses.

## 5. Conclusions

These results suggest that abnormalities in the vascular and lymphatic systems are observed from the early stage of colorectal cancer development and that VEGF-A derived from TAMs is important for tumor angiogenesis. In addition, CD34-positive lymphatic vessels observed in advanced areas of tumor invasion have not been reported in Japan, but there are reports that they are neoplastic lymphatic vessels, and further investigation is needed.

## Figures and Tables

**Figure 1 ijms-26-02919-f001:**
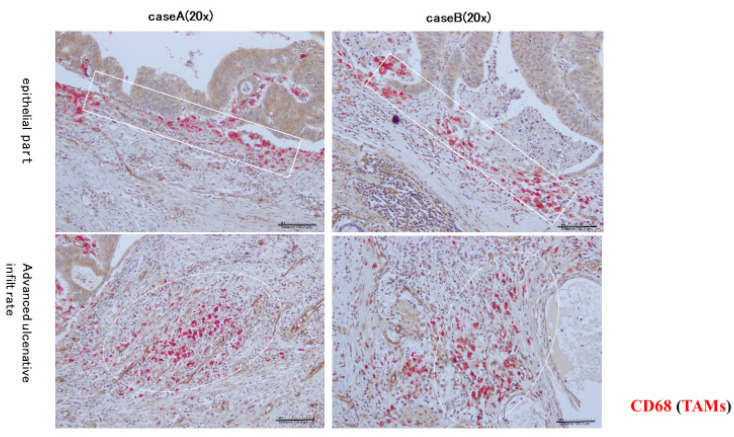
Characteristics of TAMs and high-density integration areas using CD68 antibody staining. The figures in the top row show those areas where TAMs were arranged in contact with the epithelium of the tumor gland ducts. The figures in the bottom row show the deepest infiltration zone and areas of TAM accumulation. TAMs formed a large cohesion in the epithelium of the tumor gland ducts and in the deepest invasive zone. Scale bars, 50 µm.

**Figure 2 ijms-26-02919-f002:**
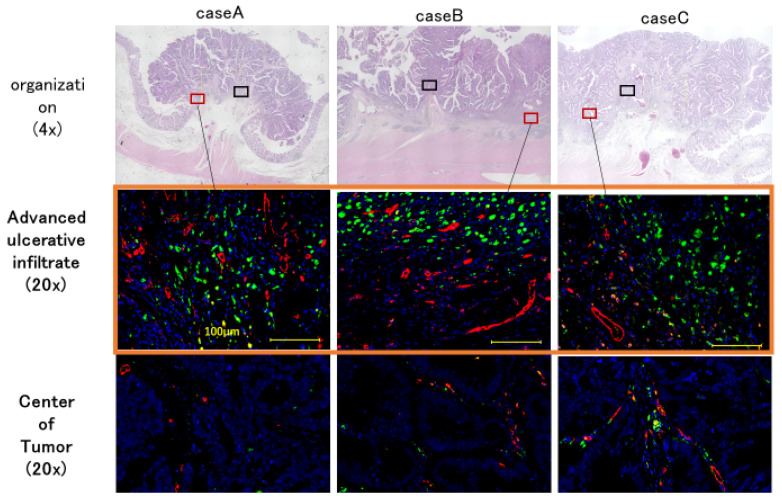
TAM and tumor vascular distribution (red, CD31; green, CD68). The heterogeneity of TAMs was assessed within tumors using early-stage colorectal cancer tissue sections from three cases subjected to fluorescent immunostaining using CD68 and CD31 antibodies. From the top, the histological images of the colorectal cancers used, the deepest tumor areas, and tumor centers are summarized. The tumor center and deepest area were also randomly selected. As a result, more TAMs and vascular endothelial cells were observed in the deepest tumor area than in the central area. Scale bars, 100 µm.

**Figure 3 ijms-26-02919-f003:**
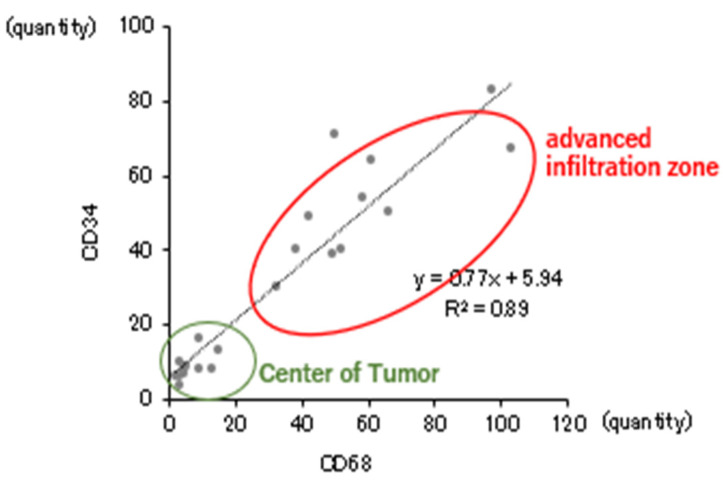
Correlation between the number of TAM and tumor micro-vessels.

**Figure 4 ijms-26-02919-f004:**
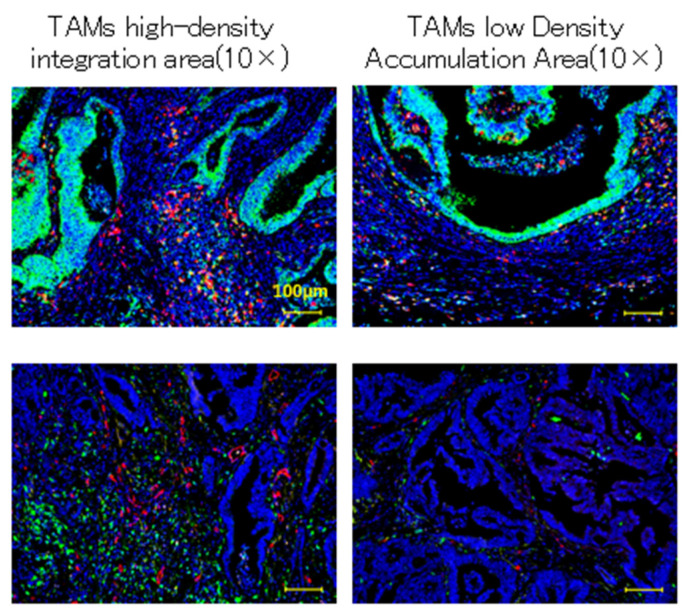
TAMs, VEGF-A, and tumor vascular characteristics. Using CD68 (red) and VEGF-A (green) antibodies, fluorescent immunostaining was used to characterize tumor vessels in areas of high- and low-density accumulation of TAMs. The figures on the left and right show the high- and low-density accumulation areas of TAMs, respectively, at 10× magnification. The results showed that cancer cells and TAMs expressed high levels of VEGF-A, and vascular density increased only around TAMs.

**Figure 5 ijms-26-02919-f005:**
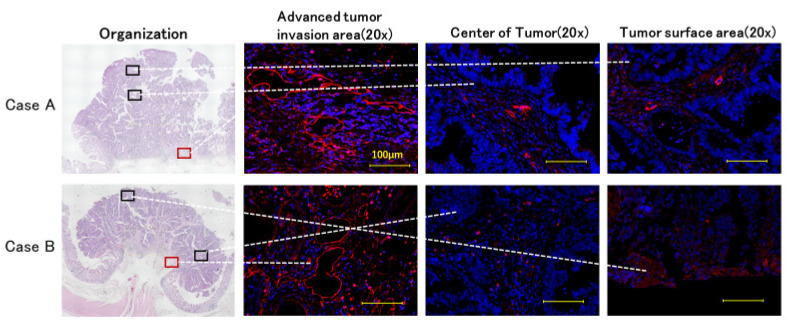
Distribution of tumor lymphatic vessels (red, podoplanin). Two cases were used to evaluate lymphatic vessels in three areas using fluorescent immunostaining: the deepest infiltration zone, tumor center, and surface area. The lymphatic vessels in the invasive deepest areas were more dilated than in the other two. The glandular vessels stained with podoplanin were defined as lymphatic vessel endothelium. This lymphatic endothelium tended to be more abundant in the more deeply infiltrated areas.

**Figure 6 ijms-26-02919-f006:**
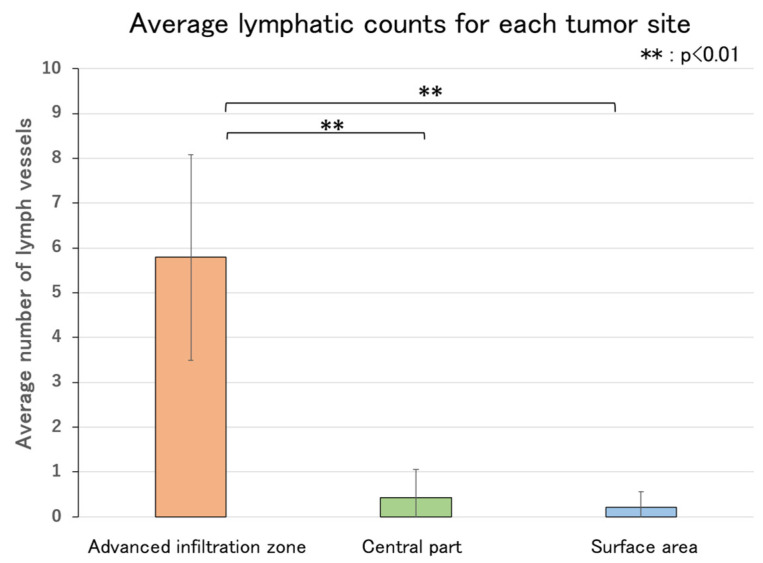
Average number of lymphatic vessels in each tumor site. The distribution of tumor lymphatic vessels was examined by measuring the number of lymphatic vessels per 1.38 × 10^5^ µm^2^ at three locations, each in the deepest infiltration, central, and surface areas.

**Figure 7 ijms-26-02919-f007:**
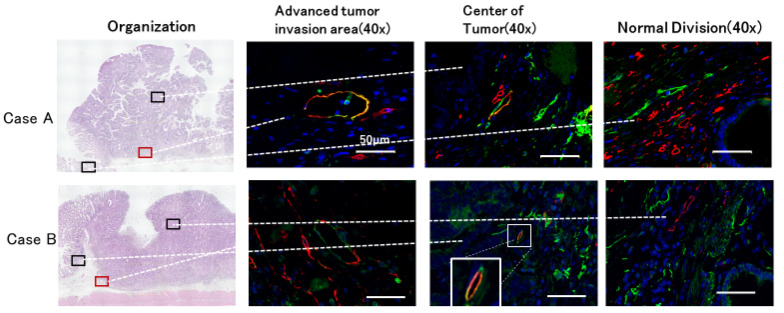
Distribution of CD34-positive lymph vessels (red, CD34; green, podoplanin). Fluorescent immunostaining with CD34 and podoplanin antibodies was performed on colorectal cancer tumors in two cases to observe lymph vessel expression. Observations were made in three locations: the deepest invasive area, the central area, and the normal area. As a result, the overlap of CD34 and podoplanin antibodies was observed in the deepest infiltrated area and the central area. The glandular vessels of the lymphatic vessels with a yellowish color due to this antibody overlap are called CD34-positive vessels. These CD34-positive vessels were rarely seen in normal areas.

**Figure 8 ijms-26-02919-f008:**
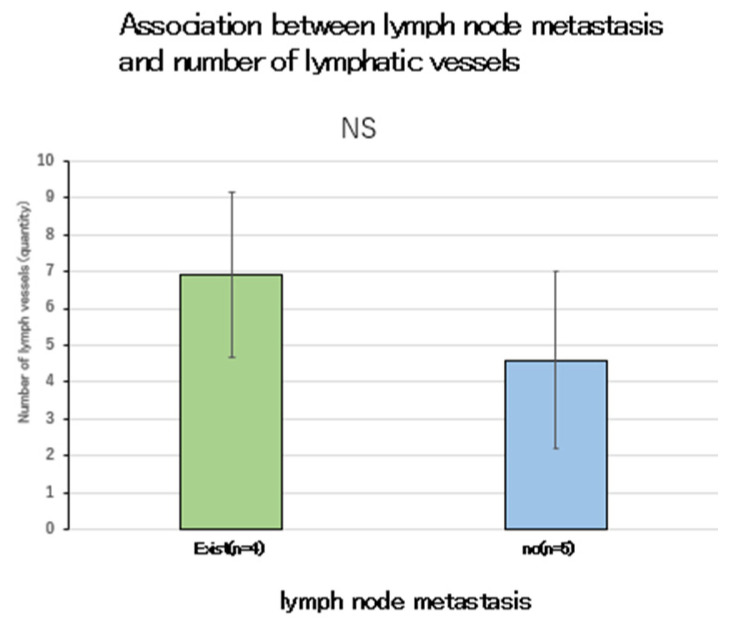
Association between lymph node metastasis and the number of lymphatic vessels. NS: not significant.

**Figure 9 ijms-26-02919-f009:**
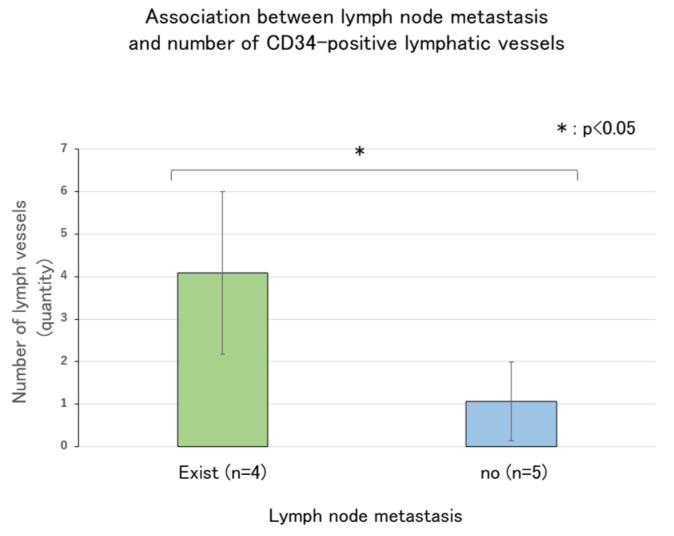
Association between lymph node metastasis and the number of CD34-positive lymphatic vessels.

**Table 1 ijms-26-02919-t001:** Antibody used.

Antibody	Animal	Company
CD31	Rabbit	BD-B (Franklin, TN, USA)
CD68	Rabbit	Cell Signaling Technology (Danvers, MA, USA)
CD34	Mouse	Nichirei Bio Science (Tokyo, Japan)
VEGF-A	Rabbit	Santa Cruz Biotechnology (Dallas, TX, USA)
Podoplanin	Mouse	Angio Bio Co. (Santiago, MN, USA)

## Data Availability

The data supporting this study’s findings are available from the corresponding author upon reasonable request.

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
