# Peer review of "Lymphangiogenesis in the Deepest Invasive Areas of Human Early-Stage Colorectal Cancer"

_ijms, 2025, doi:10.3390/ijms26072919_

Round 1
Reviewer 1 Report
Comments and Suggestions for Authors
Introduction: I believe that this section is well-structured and effectively introduces the clinical issue and its global relevance. However, considering that lymphangiogenesis is the central theme of the study, a more detailed introduction to itsmolecular mechanism could enhance that contextual framework. Additionally, the introduction should conclude with a sentence that more clearly anticipates the specific objectives of the study.
Reorganizing the order of the sections could help maintain the overall clarity of the manuscript.
Materials and methods: This section is methodologically appropriate, and the techniques used are clearly described, ensuring the reproducibility of the study. The statistical analysis is also well-structured. However, the use of only 11 samples significantly limits the statistical power of the study. It would be beneficial to discuss this limitation and provide a sample size analysis to determine the number of cases required for more generalizable results. Additionally, it is not entirely clear how the HPFs were selected within the tumor tissues and whether specific criteria were applied to minimize inter-observer variability.
Results: acceptable in their current form.
Discussion and conclusions: acceptable in their current form.
Author Response
The author has revised the manuscript in accordance with the reviewers' comments.
Reviewer 2 Report
Comments and Suggestions for Authors
The authors address an important topic about colon cancer metastasis, despite this some points should be addressed before publishing.
Abstract should be adhere standard abstract writing, background, aim, methods, results and conclusions.
Introduction should be contains information about colorectal cancer epidemiology global and regional. Moreover, add information about colorectal cancer risk factors. Please, add information about the preferred tracking for colorectal cancer metastasis. At the end of introduction section, please add the rationale and the aim of the study in a clear statements.
Results: please add the statistical data ( Mean, SD, significance, p values) to the figure legends, figures 6, ,8, and 9.
Discussion:: please support your findings with similar work and add the contradictory if present. As well, discuss the lymphatic metastasis as preferred track for cancer metastasis compared to blood metastasis. Please add the study limitations.
References: references list must be updated with addition of 2023, 2024, 2025 citation dated.
Minor issues
Please, check the manuscript for miss use of acronyms. Moreover, check the manuscript for long sentences or paragraphs without references citation.
Comments on the Quality of English Language
Please, check the manuscript for minor grammar errors and syntax.
Author Response

(The authors gave the same response as above.)
